# Arrhythmogenic Right Ventricular Cardiomyopathy and Differential Diagnosis with Diseases Mimicking Its Phenotypes

**DOI:** 10.3390/jcm11051230

**Published:** 2022-02-24

**Authors:** Nadine Molitor, Firat Duru

**Affiliations:** 1Division of Arrhythmias and Electrophysiology, Clinic for Cardiology, University Heart Center Zurich, 8091 Zurich, Switzerland; nadine.molitor@usz.ch; 2Center for Integrative Human Physiology, University of Zurich, 8057 Zurich, Switzerland

**Keywords:** arrhythmogenic cardiomyopathy, right ventricular, outflow tract tachycardia, Brugada syndrome, athlete’s heart, dilated cardiomyopathy, myocarditis

## Abstract

Arrhythmogenic right ventricular cardiomyopathy (ARVC) is an inherited heart muscle disease, which is characterized by fibro-fatty replacement of predominantly the right ventricle (RV). The disease can result in ventricular tachyarrhythmias and sudden cardiac death. Our understanding of the pathophysiology and clinical expressivity of ARVC has been continuously evolving. The diagnosis can be challenging due to its variable expressivity, incomplete penetrance and the lack of specific diagnostic criteria. Idiopathic RV outflow tract tachycardia, Brugada Syndrome, athlete’s heart, dilated cardiomyopathy, myocarditis, cardiac sarcoidosis, congenital aneurysms and diverticula may mimic clinical phenotypes of ARVC. This review aims to provide an update on the differential diagnosis of ARVC.

## 1. Introduction

Arrhythmogenic right ventricular cardiomyopathy (ARVC) is an inherited cardiomyopathy, which is characterized by cardiomyocyte loss and fibro-fatty replacement of predominantly the right ventricular (RV) myocardium. Biventricular and left ventricular (LV)-dominant disease variants have also been identified as different phenotypic expressions of this disease [1,2] The fibro-fatty alteration typically spreads from the subepicardium towards the subendocardium, predisposing patients to life-threatening ventricular arrhythmias and ventricular dysfunction [2,3,4,5].

The disease is frequently inherited as an autosomal dominant trait caused by mutations in genes encoding for desmosomal and non-desmosomal proteins. Inheritance is typically autosomal dominant with incomplete penetrance and variable expressivity [6,7]. The prevalence of ARVC has been reported to be approximately 1 in 1000 to 5000 [8,9]. ARVC is present in up to 20% of individuals, especially athletic patients, who experience life-threatening tachyarrhythmias at a young age [10,11]. The feared complication of sudden cardiac death (SCD) occurs in up to 11% as the first manifestation of disease, before structural myocardial changes are detectable [12]. Therefore, early detection and differentiation of ARVC from other diseases is crucial. 

There are several primary arrhythmic conditions, as well as structural disorders involving the ventricular myocardium, which overlap with the clinical phenotype of ARVC. Moreover, these disorders may result in ECG changes that mimic those that are observed in patients with ARVC (Table 1). 

## 2. Primary Arrhythmic Conditions

### 2.1. Idiopathic Right Ventricular Outflow Tract Tachycardia

Ventricular arrhythmias are a common finding in patients with ARVC and often represent the initial presentation of the disease. Ventricular tachycardias (VT) and premature ventricular complexes (PVCs) in ARVC typically have a left bundle branch block (LBBB) morphology, which indicates an origin from the RV. Most arrhythmias arise from the RV outflow tract (RVOT) or the subtricuspid region. The former has an inferior axis, whereas the latter shows a superior axis. When LV involvement is present, ventricular arrhythmias with a RBBB morphology may also be seen. The RVOT is also the site of origin for idiopathic RVOT-VT, a not so rare entity occurring in patients with structurally normal hearts. Differentiation between idiopathic VT from RVOT and ARVC is very important, given the benign nature of the former and the need for SCD risk stratification and family screening in the latter. 

ECG differences in sinus rhythm and during ventricular arrhythmias can be helpful in differentiating the two diseases. The presence of T wave inversion in V1–V3 in baseline sinus rhythm is present in patients with ARVC in up to 50%, but these changes may also be observed in 4% of RVOT-VT patients [13,14,15,16]. Ventricular late potentials may also differentiate ARVC patients from those with idiopathic RVOT arrhythmias. Epsilon waves are pathognomonic low-amplitude signals between the end of QRS complex and onset of the T wave in the right precordial leads and can be observed in 10–37% [14,17]. Regarding ECG findings during VT and PVC, some studies have shown longer mean QRS durations in lead I in ARVC patients [18,19]. In addition, earlier onset of QRS in lead V1, more delayed precordial transition (>V5) as well as more QRS notching have been reported in ARVC patients as compared to those with idiopathic RVOT-VT [19]. 

Idiopathic RVOT-VT is caused by focal enhanced automaticity and/or triggered activity localized to the infundibular region of the RV. The origin of VT is often septal in origin in RVOT-VT [20,21].

Patients with widespread ARVC can show several morphologies of VT. The presence of more than one form of VT should increase the pre-test probability for ARVC, as this likely reflects more diffuse pathology affecting multiple sites of the RV [21,22]. Invasive electrophysiological study seems to be helpful in differentiating the two conditions, as previous studies have described 82–93% of ARVC patients being inducible with critically timed extra stimuli vs. only 3% in the idiopathic group, indicating the re-entrant mechanism of the ventricular arrhythmias in the majority of patients with ARVC [21,22].

### 2.2. Brugada Syndrome

Although ARVC and Brugada Syndrome (BS) are distinct clinical entities with different genetic backgrounds, a phenotypic overlap can be observed with these conditions. BS is observed in patients with structurally normal hearts and has been linked to autosomal dominant mutations in SCN5A, a gene located on the short arm of chromosome 3 (p21–24) that encodes for the α subunit of the sodium channel. It is characterized by dynamic ST-segment elevations (accentuated J wave) in leads V1 to V3, which can be followed by negative T waves. RBBB pattern of varying degrees is observed in some patients. The syndrome is associated with syncope and SCD caused by polymorphic VT. An acquired form of BS has also been recognized, which may be due to a wide variety of drugs and conditions that alter the balance of currents during the early phases of the action potential [23]. 

Among patients with ARVC, there is a subpopulation with a clinical and ECG pattern similar to BS, including the presence of Type 1 ST-segment elevation and polymorphic VT. These cases of ARVC are thought to represent an early or concealed form of the disease, during which ultrastructural changes are below the resolution power of routine imaging techniques [24,25]. Provocation with sodium channel blockers may induce the characteristic coved-type ECG pattern of BS in up to 16% of patients diagnosed with ARVC [26]. On the other hand, some patients with BS may fulfil the diagnostic criteria for ARVC based on the 2010 Task Force Criteria. RV wall motion abnormalities may be observed in 71% of patients with Brugada ECG patterns and structural alterations corresponding to localized ARVC may be observed in 16% [27]. Moreover, 8% of patients may fulfil both BS and ARVC criteria [27]. ECG changes in ARVC include the presence of epsilon waves, localized prolongation of the QRS complex, QRS duration prolongation and prolonged S wave upstroke in leads V1–V3, all of which reflect activation delay in the RV [28]. Epsilon waves appear to be rare in BS patients [29], whereas other conduction abnormalities appear to be more common, with 40% of BS patients demonstrating terminal activation delay >55 ms [29]. Late potentials on signal-averaged ECG also appear to be common in BS patients [30]. The diagnostic value of the epsilon wave has been questioned in the last decade because its identification and interpretation are largely influenced by ECG filtering and sampling rate, with large interobserver variability. Hence, in the 2020 Padua Criteria, this ECG marker has been downgraded to a minor ECG depolarization criterion [28].

In ARVC, atypical RBBB morphology can be seen with a “plateau” in the R wave in lead V1. The ST segment is, at times, elevated, but does not usually mimic type 2 Brugada pattern. Moreover, T waves are more frequently negative in many precordial leads (V1 to V3–V5) and the ECG pattern is fixed with no spontaneous changes in morphology or ST elevation [31].

Molecular experimental studies suggest that overlapping phenotypes of ARVC and BS may arise from the common origin from the “connexome”, which is a protein network located at the intercalated discs where desmosomes and sodium channels work synergistically to regulate adhesion and excitability of myocytes. As a consequence of the crosstalk between desmosomes and sodium channels, the loss of expression of desmosomal proteins may cause a concomitant sodium channel dysfunction with current reduction. In this regard, it has been demonstrated that loss of expression of desmosomal proteins may cause a Brugada phenotype due to the reduction in sodium current [32].

## 3. Arrhythmias in Structural Disease

ARVC may be mimicked by arrhythmic conditions involving the right or the left heart. The athlete’s heart is an entity of the right heart and can overlap with classical early stage ARVC, whereas dilated cardiomyopathy (DCM), myocarditis and cardiac sarcoidosis are left heart diseases, which may complicate differentiation with more advanced stages of ARVC.

### 3.1. Athlete’s Heart 

Symmetric RV and LV dilatation, along with a symmetrical increase in cardiac mass and preserved systolic and diastolic ventricular function, is a well-recognized adaptive remodelling of the heart to intense physical exercise called athlete’s heart. Some features of the athlete’s heart may overlap with those of ARVC, especially during the early stages of the disease and may lead to missed diagnosis of a potentially life-threatening disease. Furthermore, athletic activity may favour ARVC disease expression and worsen its severity. According to the Padua Criteria, diagnosis of ARVC requires a combination of regional RV wall motion abnormalities and global RV dilation and dysfunction on echocardiography, cardiac MRI or angiography. Cardiac chamber volumes are more accurately measured by the modern CMR cine technique with steady-state free procession images, which provide a superior contrast between blood and endocardium at the endocardial border with less blood flow dependence. In order to increase the diagnostic accuracy of CMR imaging findings, the Padua Criteria recommend using reference values for RV cavity size and systolic function, normalized for age, sex, and body surface area (BSA), according to current nomograms provided by international societies of cardiovascular imaging [28]. Physical exercise acutely and transiently increases pressure, afterload, and wall stress to a disproportionately greater extent in the RV than in the LV, and therefore, prolonged and intense training can induce an increase in RV cavity size and mass [33,34,35] and often exceeds the normative reference values proposed for the general population [36]. For these reasons, a relevant proportion of competitive athletes may fulfil the minor and major echocardiographic dimensional criteria for the diagnosis of ARVC, especially in the 2010 Revised International Task Force Criteria [37]. Athletes engaged in combined disciplines (e.g., rowers, canoeists) have the highest degree of RV remodelling, whereas athletes who are predominantly involved in strength exercise have the lowest [36]. 

The normal reference value of 100 mL/m^2^ (men) and 90 mL/m^2^ (women) of RV end-diastolic volume/BSA proposed by the Task Force Criteria may lack specificity for ACM, especially due to the overlap with physiologic adaptive changes in the athlete’s heart, which can produce an increase in both LV and RV volumes that is well beyond the upper limit of normality reported in the general population. In this regard, proper reference values for RV volume in the athlete’s heart are currently available and recommended by the Padua Criteria for differential diagnosis of physiologic versus pathologic RV dilatation, especially for endurance sports such as rowing or canoeing, associated with the greatest RV dimensional remodelling [28].

RV enlargement in athletes is usually accompanied by remodelling of LV Function [38], reflecting a global and symmetrical adaptation of the heart to the hemodynamic changes induced by training. ARVC patients usually show no or mild LV dilation [39,40]. A RV/LV ratio <0.9 has been proposed to distinguish between RV physiological remodelling and ARVC [40].

Despite marked RV dimensional remodelling, systolic and diastolic functions are usually normal in the athlete’s heart [36,37,38], although a lower reference value than the general population of RV fractional area change (FAC) of 32% has been recently proposed for athletes [36]. A reduction of RV strain is confined to the basal segment, particularly for endurance athletes. This functional adaptation is likely a physiological consequence of exercise-induced hemodynamic changes [41]. Recurrent dilatation of cardiac chambers during intense physical exercise, stimulating resident fibroblasts and macrophages, may result in the deposition of collagen with patchy areas of fibrosis, a potential substrate for re-entrant ventricular arrhythmias [42,43].

Cardiac magnetic resonance (CMR) tissue characterization analyses in highly trained athletes have failed to show the presence of Late Gadolinium Enhancement (LGE) macroscopic fibrosis of the RV wall, with the exception of “junctional” LGE confined to the interventricular septum in the vicinity of the RV attachment, which is considered a normal feature of the athlete’s heart [44,45]. This is in contrast to the LGE pattern of right dominant ARVC, which is typically located in the free wall, subtricuspid region and outflow tract, with a nonischaemic subepicardial or midmyocardial distribution [46].

T-wave inversion in V1–V4 in individuals with complete pubertal development is a frequent ECG marker of ARVC, associated with life-threatening ventricular arrhythmias [47,48], although the extent and distribution of T-wave inversion across all precordial leads has limited value for differential diagnosis for the athlete’s heart, because physiological cardiac adaptation to regular exercise may also include T-wave inversion in adults. On the other hand, T-wave inversion in the adult athlete’s ECG is usually preceded by J-point/ST-segment elevation and represents a variant of training-related anterior early repolarization. J-point elevation preceding T-wave inversion has been rarely observed in the total ARVC cohort and not found in the subgroup of athletes with ARVC, markedly increasing the specificity of ARVC diagnosis [49]. Inferior T-wave inversions are also common in ARVC patients [50]. Moreover, the ECGs of athletes do not exhibit depolarization abnormalities such as right precordial QRS prolongation with delayed S wave upstroke, increased terminal activation delay or QRS fractionation, which are commonly seen in ARVC as a reflection of delayed activation through diseased RV free wall. ARVC patients show more often Q waves or lower precordial QRS amplitudes (<1.8 mV), more than 1000 PVCs and more than 500 non-RVOT PVCs per 24 h, as well as ventricular tachyarrhythmias and attenuated blood pressure response during exercise, compared to subjects with athletic hearts [40]. 

Some echocardiographic features have also been identified in order to help to distinguish between physiological and pathological RV remodelling. A relevant echocardiographic difference between ARVC patients and competitive athletes is the presence of RV wall motion abnormalities. RV bulging, dyskinesia, akinesia, and aneurysms, which are typical findings of ARVC, are not observed in healthy athletes [38,40,51]. Competitive athletes have RVOT diameters beyond the cut-off values, fulfilling minor or major criteria for the diagnosis of ARVC [36,38]. On the other hand, RV dilation in healthy athletic hearts is more pronounced in RV inflow than in RVOT [51], while ARVC patients without severe RV dilation or dysfunction often have both RV inflow and outflow tract dilation [39,52]. ARVC patients present with significantly larger RA compared to athletes, resulting in a greater RAVI/LAVI [53].

### 3.2. Dilated Cardiomyopathy (DCM)

DCM and ARVC are clinically heterogeneous diseases of the myocardium, which are both associated with mechanical and electrical dysfunction. The aetiology includes genetic and non-genetic causes (e.g., drugs, toxic agents, endocrine disorders, nutritional deficiency, infection). While ARVC is clinically characterized by ventricular arrhythmias, predominantly from the RV, often preceding structural changes, DCM is characterized by LV contractile dysfunction and progressive heart failure, with arrhythmias often being less prominent [54]. RV dysfunction may also be present in DCM, representing a strong predictor for worse clinical outcome [55].

Classical ARVC initially shows signs of RV involvement. However, histopathological and functional LV involvement is present in 76–84% of these patients and left-dominant forms also exist [2,4]. Compared to DCM, patients with left-dominant ARVC often have significant ventricular arrhythmias, disproportionate to the morphological abnormalities and impaired LV systolic function [2]. Regarding myocardial fibrosis, non-ischemic myocardial fibrosis, as evidenced by post-contrast CMR sequences, is found in approximately 30–50% of patients of DCM in the form of patchy mid-myocardial LGE, the amount of which is unrelated to the severity of LV systolic dysfunction, although it is an independent predictor of malignant arrhythmias [56,57,58]. LV involvement in ARVC may manifest as sub-epicardial or mid-myocardial LGE, often involving the inferior and lateral walls without concomitant wall motion abnormalities [2,46]. Septal LGE is present in more than 50% of cases with left dominant ARVC [46]. In addition, LV fatty infiltration has been shown to be a prevalent finding in ARVC, often involving the subepicardial lateral LV and resulting in myocardial wall thinning [59]. The principal clinical discriminating feature of left-dominant arrhythmogenic cardiomyopathy (ACM) from DCM would be the predisposition to ventricular arrhythmias in early stages of the disease, disproportionate to the morphological abnormalities and impaired systolic function [2,46].

Besides the clinical similarities, a considerable genetic overlap has also been recognized [60]. In addition to desmosomal mutations, mutations in non-desmosomal genes have been identified in ARVC. These non-desmosomal genes include, among others, desmin, titin, lamin A/C, and phospholamban, which may be mutated also in subjects with DCM [61,62,63]. A phospholamban (PLN) R14 del founder mutation has also been described in a substantial number of patients clinically diagnosed with DCM or ARVC. Those diagnosed with DCM showed an arrhythmogenic phenotype and abnormal repolarization in 59% of patients. Moreover, 17 of 39 patients with PLN mutation fulfilled a possible or borderline diagnosis of ARVC at the time of DCM diagnosis. RV dilatation was present in 15 (38%) of these patients [60].

### 3.3. Myocarditis

Differential diagnosis between ARVC and myocarditis may be particularly challenging because some patients with ARVC can present with bouts of acute or subacute myocarditis [64,65]. ARVC patients may experience episodes of acute symptoms, including chest pain and palpitations, as well as troponin release and myocardial oedema on CMR, mimicking acute myocarditis. These “hot phases” may occur as first manifestation or later during the clinical course as an expression of the myocardial disease progression [66].

On CMR imaging, LV involvement in ARVC may manifest as sub-epicardial or mid-myocardial LGE, often involving the inferior and lateral walls without concomitant wall motion abnormalities [2,46]. T2-weighted CMR imaging may detect tissue oedema in acute myocarditis. In addition, fast spin-echo T1-weighted images during the first minutes after contrast injection may be useful to detect myocardial hyperaemia and muscular inflammation suggestive of myocarditis [67,68]. However, desmoplakin cardiomyopathy, during hot phases, may also be characterized by hyperintense myocardial areas at T2-weighted CMR imaging similar to myocarditis [69].

Histologic examination may also show overlapping features because ARVC hearts may often reveal prominent myocardial inflammation. Inflammatory infiltrates may be present in 60–88% of hearts of ARVC patients, which are also associated with life-threatening ventricular tachyarrhythmias [3,4,70]. 

As in the case for myocarditis, positron emission tomography (PET) scanning in ARVC patients may demonstrate 18F-fluorodeoxyglucose (FDG) uptake abnormalities. FDG uptake has been found in the LV myocardium in all cases, while RV uptake has also been observed [71]. Differential diagnosis between acute myocarditis or post-myocarditis scar and ARVC requires accurate medical history, clinical family screening, careful analysis of ECG and imaging abnormalities, and molecular genetic testing.

### 3.4. Cardiac Sarcoidosis

Sarcoidosis is a systematic inflammatory disease characterized by the formation of non-caseating granulomas. Cardiac involvement (cardiac sarcoidosis, CS) has been reported in up to 40% of cases. CS is usually diagnosed in patients who have multisystem disease. Isolated CS is rare, but more frequently reported in recent years with the increasing use of CMR imaging. Autopsy studies show that both the free wall and the septum of the LV are generally affected by granulomatous infiltration and fibrosis. The RV free wall may also be involved in some cases [72].

Accurate differentiation between these two conditions has implications for immunosuppressive therapy and familial screening. CS shares several clinical and morphological features with genetically determined ARVC. Both diseases can present with ECG repolarization or depolarization abnormalities, structural cardiac abnormalities, or arrhythmias. Increasing numbers of case reports have described patients initially suspected of ARVC diagnosed using the current criteria who were later found to have biopsy-proven CS [73,74]. In a prospective cohort of 20 patients with suspected ARVC, 16 fulfilled definite diagnosis according to the ARVC Task Force Criteria. However, three of these patients (15%) were later diagnosed with CS after noncaseating granulomas were found on cardiac biopsy [74].

In general, compared to ARVC, patients with CS are older at symptom onset, more likely to have comorbidities and develop heart failure symptoms over time [75,76]. Moreover, a family history of SCD strongly distinguishes these two diseases, since no patient in the CS group reported a history of familial SCD. CS patients have lower LVEF and increased LV diastolic and systolic volumes, whereas patients with ARVC have lower RVEF and increased RV diastolic and systolic volumes. Isolated LV involvement has been demonstrated only in those with CS. Most patients with ARVC have RV involvement, whereas only 48% of those with CS have RV involvement. The presence of mediastinal lymphadenopathy has been reported only in the CS group (68%). CMR imaging has shown LGE in 73% of CS patients and 19% in the ARVC group. Moreover, scar locations are different between the two cohorts. For example, isolated LV involvement has been seen only in the CS group. Forty-three percent of the CS group and none in the ARVC group demonstrated intraventricular septum dysfunction or septal LGE. In addition, the presence of short-tau inversion recovery positivity was found more often in the CS group, likely indicating that some patients with CS had an acute or subacute inflammatory response. MRI findings demonstrating the presence of septal involvement, isolated LV infiltration and/or mediastinal lymphadenopathy favours the diagnosis of CS [76]. Regional FDG uptake in PET indicates active inflammatory lesions, which should raise the suspicion of CS [77]. However, this finding is not specific for CS and may also be found in ARVC [71].

On surface ECG, PR interval prolongation and high-grade atrioventricular (AV) block are exclusively associated with CS [75]. The presence of infra-Hisian conduction abnormalities such as bundle branch block and AV block is a consequence of the predilection of granulomatous infiltration for the basal interventricular septum, which is usually absent in ARVC [75].

Regarding electrophysiological characteristics, patients with CS have mappable VTs originating most commonly from the apical region [78], whereas in patients with ARVC, most VTs originated from the RVOT or the subtricuspid region [78,79]. Significantly more monomorphic VT morphologies may be inducible in patients with CS [78].

### 3.5. Congenital Ventricular Outpouchings (Aneurysm/Diverticula)

Ventricular outpouchings include acquired pathologies comprising ventricular aneurysms (or pseudoaneurysms) and congenital diverticula. Ventricular aneurysms and diverticula are rare congenital heart defects due to a segmental developmental anomaly of the LV wall, and rarely of the RV wall. The two entities can be distinguished on the basis of the histologic features of the outpouching, i.e., the absence of myocardium in aneurysms and the presence of myocardium in diverticula [80,81]. Aneurysms appear on cine CMR or angiography as large akinetic areas with paradoxical systolic motion and dyskinetic pouches with a wide connection to the ventricle. LGE sequences offer the possibility to characterize the wall tissue with demonstration of either endocardial or transmural myocardial-replacement fibrosis. The walls of diverticula show a preserved, synchronal systolic contractility in consistence with normally developed muscular layers [82]. Based on their localization, they may be identified as apical or non-apical ventricular diverticula, the latter being seen more frequently [83]. Congenital apical diverticulum is a part of the Cantrell Syndrome and has a common defect in embryologic development with midline thoracoabdominal formation. The mechanism may result from a failure of normal fusion of the paired primitive mesoderm in combination with abnormal fusion of the cardiac loop to the yolk sac before it descends, leading to the formation of muscular apical diverticula [84]. While the aetiology of non-apical aneurysms and diverticula has been attributed to an abnormality in embryogenesis, ventricular aneurysms may also be acquired during the prenatal period, potentially as a result of a viral infection or coronary lesions, including stenosis, hypoplasia, and localized intimal proliferation [85,86,87].

Idiopathic LV aneurysms, presumably of congenital aetiology, may cause life-threatening ventricular arrhythmias [88]. Therefore, differential diagnosis between left-dominant ARVC and idiopathic LV aneurysms/diverticula can be challenging. While congenital ventricular aneurysms may be associated with adverse outcomes, the prognosis for ventricular diverticula is often good [83].

## 4. Conclusions

ARVC is a rare but important cause of potentially life-threatening ventricular arrhythmias in the young and athletes. Although significant advances regarding our understanding of its pathophysiology have been made, ARVC diagnosis is still particularly challenging due to the lack of unique specific diagnostic criteria, incomplete penetrance, and the wide spectrum of clinical manifestations, from concealed forms to focal RV involvement, to biventricular or left dominant cardiomyopathy. Due to considerable clinical overlap with other diseases, both of primarily arrhythmogenic and of structural origin, each individual case needs to be examined carefully, taking into account the family history, clinical presentation, ECG changes and imaging manifestations, as well as the genetic background.

## Figures and Tables

**Table 1 jcm-11-01230-t001:** Arrhythmogenic Right Ventricular Cardiomyopathy and Differential Diagnosis with Diseases Mimicking its Phenotypes. Abbreviations: DPT = Delayed precordial transition, LAD = Left axis deviation, LBBB = Left Bundle Branch Block, LP = Late potential, N = Normal, RBBB = Right Bundle Branch Block, RVOT = Right ventricular outflow tract, TAD = Terminal activation delay, TWI: T-Wave Inversion, VF Ventricular Fibrillation, VT = Ventricular Tachycardia, WMD = Wall Motion abnormalities.

			ARVC	RVOT	Brugada	Athlete’s Heart	Cardiac Sarcoidosis	DCM	Myocarditis	Congenital Aneurysm
ECG	QRS voltages		Low or N	N	N	Increased	Low or N	N or increased	N	N
	QRS Complex		QRS prolongation/V1–3 delayed S upstroke/TAD/ LP	N	RBBB pattern LAD	Incomplete RBBB	Infra-Hissian conduction abnormalities	LBBB, LAD	N or prolonged	−
	Repolarization		TWI V1–3 (+/− V4/5/6) +/− II, III, aVF	Rarely TWI V1–3	ST elevation + TWI V1–3	Early Repol. or TWI V1–3 preceded by J−point elevation	Prolonged QTc dispersion, Tpe and Tpe/QT ratio	Strain pattern of ST−segment	TWI or ST−elevation	Sometimes TWI esp. V4–6 (apical location)
	Epsilon waves		Possible	No	Rare	No	No	No	No	No
VTs	Mechanism		Scar related Re-entry	Enhanced automaticity/ triggered activity	Phase 2 re−entry or local micro−re-entry	No VTs	Scar related Re-entry	Scar related Re-entry/functional	Inflammation (acute)/ Scar related Re-entry	Scar related Re-entry
	Origin		Non−septal RVOT/ Tricuspid annulus	Anteroseptal RVOT	RV epicard/ RVOT	No VTs	Ventricular apex, basal septum	LV Intra−mural/septum	LV	Aneurysm location
	Morphology		Monomorphic, possible several VTs	Monomorphic	Polymorphic VT, VF	No VTs	Monomorphic, several VTs	Monomorphic, several VTs	Polymorphic (acute) or monomorphic	Monomorphic
	Axis and QRS speciality/ BBB pattern		LBBB + superior/inferior axis or RBBB or VF, QRS in I > 120 ms QRS−notching, Earliest QRS in V1/ DPT	LBBB + inferior axis Usually, QRS in I <120 ms	Polymorphic or LBBB + inferior QRS axis	No VTs	LBBB or RBBB	RBBB	RBBB	RBBB or LBBB
Structural changes (TTE or CMR)	RV	Volume	Dilated mainly RVOT+/−subtricuspid aneurysm	N	N (or mild RVOT dilatation)	Dilated mainly ventricle	N (may be dilated in PH)	N (dilated in end stage)	N	N (Only local *)
		Regional WMD	Yes	No	No (possible)	No	Possible	No	Possible	(Local WMD) *
	RVEF		Reduced	N	N	N or mild reduced	N or reduced	Reduced in progressive LV dysfunction	N	May be reduced *
	LV	Volume	N	N	N	N	N or dilated	Dilated	N	N (Only local °)
		Regional WMD	Hypokinetic (left dominant form) +/− regional WMD	No	No	No	Hypokinetic +/− regional WMD (septum + thinning)	Hypokinetic no regional WMD	Hypokinetic +/− regional WMD	(Local WMD) °
	LVEF		May be reduced	N	N	N	Reduced	Reduced	May be reduced	May be reduced °
	RV/LV		>1	<1	<1	<1	<1	<1	<1	−
CMR LGE	Myocardial Layer		Subepi/midmyo	No	No	No	Midmyo	Midmyo	Subepi	−
	RV Location		Lateral + wall thinning/subtricuspid region/RVOT	No	No	No or junctional	RV free wall	No	Rarely involved	Esp. apical + wall thinning *
	LV Location		Inferolateral wall/ Inferior/ septal junction	No	No	No	Basal segments esp. of the septum + free wall	Patchy often involving the septum	Lateral wall or lateral + septal or diffuse	Esp. apical + wall thinning °

* Right sided aneurysm ° left sided aneurysm.

## Data Availability

Not available for this type of article.

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
