# Peer review of "Arrhythmogenic Right Ventricular Cardiomyopathy and Differential Diagnosis with Diseases Mimicking Its Phenotypes"

_jcm, 2022, doi:10.3390/jcm11051230_

Round 1

Reviewer 1 Report

The authors provide an update of the data concerning the differential diagnosis of ARVC and alternative diagnoses by reviewing the following nosological entities one by one

Idiopathic RV outflow tract tachycardia, Brugada Syndrome, athlete's 15heart, dilated cardiomyopathy, myocarditis, cardiac sarcoidosis, congenital aneurysms

General comments

I must congratulate the authors on the detailed vision and the attempt to be exhaustive both in terms of electrical changes in depolarisation or repolarisation or in terms of structural changes or presence in LGE an MRI

However, the new 2020 criteria are not mentioned

Evolving Diagnostic Criteria for Arrhythmogenic Cardiomyopathy.

Corrado D, et al. J Am Heart Assoc. 2021

Especially since criteria such as the epsilon wave or the late potentials   has been downgraded to a minor ECG depolarization criterion and late potentials on signal-averaged ECG, which was a minor criterion in the 2010 TF criteria, is no longer included among the 2020 International criteria

the authors state

“According to the 2010 Modified Task Force Criteria, diagnosis of ARVC requires a 134 combination of regional RV wall motion abnormalities and global RV dilation and dys-135 function on echocardiography. Cutoff values for these measurements were derived from 136 studies in the general population”

however these criteria have been updated: Padua Criteria with increased modified threshold values taking into consideration the modern CMR cine technique with steady‐state free procession images, which provide superior contrast between blood and endocardium at the endocardial border with less blood flow dependence

the authors include recent data concerning the left-sided type of arrhythmogenic cardiomyopathy and also the potentially inflammatory aspects mimicking myocarditis by pointing out the recent notions of a considerable genetic overlap between AR(L)VC and DCM

Minor comments

Table 1 would benefit from being more readable and concise

fig 1 is difficult to follow

How much additional interest does Fig1 bring to the text when the recommendation in ARVC is to apply multiple criteria Structural ECG Tissue Rhythmic Family and Genetic , as the authors state in their conclusion

Reviewer 2 Report

The topic is of great interest, however I have some comments:

1) If the involvement of the left ventricle in the ARVC is about 70-80% I would drop the sentence:"A septal origin of VT is significantly 69 more often observed in idiopathic RVOT-VT than in ARVC, which usually affects the RV 70 free wall and spares the septum ."

2) I would emphasize in adults with T wave inversion in athlete’s heart section

3) This sentence is not correct “T2-weighted CMR imaging may detect tissue edema in 241 acute myocarditis, which is usually absent in ARVC”: desmoplakin cardiomyopathy, during hot phases, may be characterized by hyperintensive myocardial areas at T2W CMR similar to myocarditis (Smith DE 2020, Bariani R 2021)

4) the figure 1 in the absence of  the inclusion of CMR role, and also without the Holter EKG monitoring / exercise test represents a confusing message, therefore I would modify/drop it
